# Assessing the Trophic Impact of Bleaching: The Model Pair *Berghia stephanieae*/*Exaiptasia diaphana*

**DOI:** 10.3390/ani13020291

**Published:** 2023-01-14

**Authors:** Ruben X. G. Silva, Diana Madeira, Paulo Cartaxana, Ricardo Calado

**Affiliations:** ECOMARE & Centre for Environmental and Marine Studies (CESAM), Department of Biology, University of Aveiro, 3810-193 Aveiro, Portugal

**Keywords:** anemone, photosymbiosis, sea slug, transgenerational effects

## Abstract

**Simple Summary:**

Climate change has made bleaching events increasingly common in coral reefs. As such, scientists are rushing to better understand what will be the outcomes of bleaching on these tropical ecosystems. However, scientific focus is mostly being given to the direct impacts promoted by these events (e.g., mass mortality of corals, coral reef decay, community transitions), with indirect ones remaining largely overlooked (e.g., impacts on trophic interactions of highly specialized animals). Organisms directly affected by bleaching display a poorer nutritional value, but the effects this may have on their predators remain largely overlooked. To address this gap in scientific research, we advocate the use of the model predator–prey pair featuring the nudibranch sea slug *Berghia stephanieae* and its prey, the photosymbiotic glass anemone *Exaiptasia diaphana*. As the two species are already used as models in other research fields, one can build upon existing knowledge to determine if and how this highly specialized sea slug, that only preys upon glass anemones, is affected in terms of survival and reproductive fitness when only bleached prey are available. Moreover, this model pair will also allow investigating if such trophic effects cascade over consecutive generations, shaping populations and ultimately ruling species survival.

**Abstract:**

Bleaching events associated with climate change are increasing worldwide, being a major threat to tropical coral reefs. Nonetheless, the indirect impacts promoted by the bleaching of organisms hosting photosynthetic endosymbionts, such as those impacting trophic interactions, have received considerably less attention by the scientific community. Bleaching significantly affects the nutritional quality of bleached organisms. The consequences promoted by such shifts remain largely overlooked, namely on specialized predators that have evolved to prey upon organisms hosting photosynthetic endosymbionts and benefit nutritionally, either directly or indirectly, from the available pool of photosynthates. In the present study, we advocate the use of the model predator–prey pair featuring the stenophagous nudibranch sea slug *Berghia stephanieae* that preys upon the photosymbiotic glass anemone *Exaiptasia diaphana* to study the impacts of bleaching on trophic interactions. These model organisms are already used in other research fields, and one may benefit from knowledge available on their physiology, omics, and culture protocols under controlled laboratory conditions. Moreover, *B. stephanieae* can thrive on either photosymbiotic or aposymbiotic (bleached) glass anemones, which can be easily maintained over long periods in the laboratory (unlike photosymbiotic corals). As such, one can investigate if and how nutritional shifts induced by bleaching impact highly specialized predators (stenophagous species), as well as if and how such effects cascade over consecutive generations. Overall, by using this model predator–prey pair one can start to truly unravel the trophic effects of bleaching events impacting coral reef communities, as well as their prevalence over time.

## 1. Introduction

Global sea surface temperature has been rising since the beginning of the 20th century [1]. Data available shows that this increase has been of 0.88 °C up until 2021 [1,2,3]. Future rise in average global sea surface temperature can range between +0.86 up to +2.89 °C by 2100, according to the Shared Socioeconomic Pathways (SSPs) scenarios developed by the Intergovernmental Panel on Climate Change (IPCC) Sixth Assessment Report [1]. Under current and future climate scenarios, extreme weather events, such as marine heatwaves, are also predicted to escalate, with serious ecological and economic impacts already being reported worldwide [4,5].

The rise in ocean temperature is one of the main drivers of bleaching, an event on which the symbiosis between animals and their photosynthetic endosymbionts is disrupted, resulting in the loss of the animal’s pigmentation [6,7]. Under all Representative Concentration Pathways (RCP), annual bleaching events are predicted to occur on more than 90% of world’s coral reefs [8]. Bleaching leads to the loss of coral coverage, which will ultimately affect predator–prey interactions amongst organisms in these ecosystems. The lack of coral coverage and absence of hiding places for preys will enhance predatorial pressure for coral dwelling animals [9]. Hence, bleaching promotes a major impact throughout coral reef food webs [10]. For example, coralivorous fish, such as wrasses and butterflyfish, where found to enhance their consumption of coral polyps right after bleaching events, but halted their consumption five to seven days later [11]. Another trophic impact already reported during bleaching events is the dramatic shift in the nutritional value of bleached organisms, as these loose access to the pool of photosynthates produced by their photosynthetic endosymbionts [7]. One of the most dramatic nutritional shifts during bleaching can be perceived in the fatty acid composition of cnidarians hosting photosynthetic endosymbionts [12], namely the decrease in highly unsaturated fatty acids, such as eicosapentaenoic acid (EPA, 20:5*n*-3) and docosahexaenoic acid (DHA, 22:6*n*-3). The effects of such shifts may eventually cascade over the trophic web and impact other organisms, namely those that have evolved to be stenophagous predators and specifically prey on organisms hosting photosynthetic endosymbionts [13].

Stenophagous predators can be defined as organisms thriving on highly specialized diets, with some authors attributing this designation to species which prey upon one to three different species [14]. As already referred above, this specialized feeding strategy may become a serious constraint for the performance and fitness of species preying upon organisms prone to bleaching. Even if these species are still able to feed upon bleached prey, the nutritional value derived from them will always be poorer [11,13]. Due to their specialized feeding regime, maternal provisioning and reproductive success of stenophagous animals can be significantly impacted under climate change and suboptimal feeding scenarios [15]. One of such organisms, whose linkage between suboptimal trophic scenarios promoted by bleaching events and reproductive success has already started to be addressed [13], is the nudibranch *Berghia stephanieae* (Á. Valdés, 2005). This stenophagous sea slug feeds upon the glass anemone *Exaiptasia diaphana* (Rapp, 1829) (see video on supplementary material), a cnidarian that hosts endosymbiotic photosynthetic Symbiodiniaceae dinoflagellates. When fed with bleached glass anemones, adult *B. stephanieae* produces egg masses with lower levels of palmitic acid (16:0) that is catabolized to fuel embryogenesis, as well as lower levels of the essential fatty acid DHA, which is a precursor of important biomolecules associated with the ontogenetic development of the neuronal system [13].

*Berghia stephanieae* is able to retain the photosynthetic endosymbionts of its cnidarian prey, although it does not establish a true mutualistic relationship with these unicellular organisms [16]. The prevalence of this retention is now well studied [16] and non-invasive and non-destructive methods to monitor this association have also been established [17]. On the other hand, in 2020, the species upon which this sea slug feeds, the glass anemone *E. diaphana*, was presented as a model species to study cnidarian–microbiome symbiosis due to its hardiness, easiness of rearing and ability to reproduce both sexually and asexually [18]. Most importantly, this anemone, much like photosymbiotic corals, displays the ability to maintain active symbiosis with various prokaryotes, including Symbiodiniaceae, with the added benefit of being able to endure bleaching events in an aposymbiotic condition much more easily than corals [18].

While bleaching events are becoming more frequent and intense worldwide, we only now start to unravel the trophic effects that may arise from them and how these can cascade over food webs [19]. The present work advocates the use of *B. stephanieae*/*E. diaphana* as a model predator–prey pair to advance our knowledge on the trophic impacts of bleaching on stenophagous organisms feeding upon species hosting photosynthetic endosymbionts. Moreover, this model predator–prey pair will also allow to gain new insights on potential transgenerational effects promoted by the trophic impacts of bleaching on both predator and prey.

## 2. Impacts of Bleaching on Trophic Interactions in Coral Reef Ecosystems

Tropical coral reefs are some of the most diverse and productive marine ecosystems in the world, despite being mostly located in oligotrophic waters [20]. These ecosystems are amongst the most vulnerable to climate change in the marine realm, experiencing an unprecedented decline worldwide, mainly due to a phenomenon known as bleaching, whose frequency and intensity has been increasing over recent decades [19,21,22]. Bleaching, the loss of Symbiodineacea microalgae [23], is one of the main impacts of climate change in tropical coral reefs; this response to stress, mainly caused by high water temperatures, results in the disruption of the symbiosis between a host and its Symbiodiniaceae partners, which may ultimately lead to death, at least for some coral species [6,7,19].

Trophic interactions are known to shape the structure and function of communities across land and sea [24,25,26]. Climate change can modify trophic interactions by impacting species distribution and abundance, species traits (e.g., morphology and physiology), foraging habits and nutritional value of prey or by inducing environmental tolerance mismatches between predators and prey [27,28,29,30]. Coral reefs, with their complex tridimensional structure, are paramount in predator–prey interactions, as they are used by smaller fishes as hiding places [31]. Coker and colleagues [9] postulated that, with habitat degradation resulting from coral bleaching, fish will be more susceptible to predation.

Even though corals are at the mediatic forefront when it comes to bleaching, they are certainly not the only group of cnidarians being negatively impacted by these events [23]. Anemones are another group of cnidarians which are prone to bleaching. A study performed on the southern Great Barrier Reef (Australia) addressing the clownfish *Amphiprion akindynos* Allen, 1972 and its anemone partner *Radianthus crispa* (Hemprich & Ehrenberg in Ehrenberg, 1834), revealed that the abundance of clownfish was reduced up to 70% when bleached anemones were compared with conspecifics still hosting their photosynthetic endosymbionts. The same study found that *A. akindynos* loses its innate threat response when the anemone it lives on bleaches, thus not seeking shelter and, consequently, being more vulnerable to predation [32].

The impact of bleaching was also studied for animals directly predating on species hosting photosynthetic endosymbionts (namely dinoflagellates), such as corallivorous fish. By artificially inducing bleaching, *in situ*, in Kimbe Bay (Papua New Guinea), researchers observed that when the wrasse *Labrichthys unilineatus* (Guichenot, 1847) and the butterfly fish *Chaetodon baronessa* Cuvier, 1829 were presented with both healthy and bleaching corals, the two fish species preferentially predated stressed corals [11]. This same study also allowed to reveal that this feeding preference was short-lived and ceased once corals were fully bleached. In another study, bleached specimens of the soft coral *Sclerophytum maximum* (Verseveldt, 1971) decreased their concentration of both lipids and a defensive metabolite (pukalide), a feature that made them more vulnerable to predation by the pufferfish *Canthigaster solandri* (Richardson, 1845) by a factor of almost 4:1 when compared to healthy conspecific corals [33]. However, not only fish predate on corals, as there are also coralivorous invertebrates that do so, such as the sea star *Acanthaster planci* (Linnaeus, 1758), popularly known as crown-of-thorns. A study following the population of this sea star in Indonesia reported that during the bleaching event of 2016 this sea star was deprived of its favorite prey (stony corals of genera *Porites* Link, 1807, *Favites* Link, 1807 and *Pavona* Lamarck, 1801), but was able to adapt its diet according to coral species availability, thus being able to endure this extreme event [34].

Overall, the cascading impacts of bleaching in trophic interactions taking place in coral reefs are still poorly understood and, to the best of our knowledge, model organisms are yet to be presented by the scientific community to better understand these complex relationships. The focus of most research efforts has been significantly biased towards either fish or reef forming photosynthetic coral species, largely overlooking groups of highly specialized marine invertebrates, such as nudibranchs (Mollusca, Gastropoda), that may be negatively affected by the bleaching of their prey. Moreover, given the shorter lifecycles commonly displayed by these marine invertebrates, deleterious effects promoted by trophic interactions with bleached prey may well propagate across generations and ultimately shape populations and/or determine the survival of highly specialized (e.g., stenophagous) coral reef species.

## 3. *Berghia stephanieae* and *Exaiptasia diaphana*—a Model Pair to Study the Trophic Impacts of Bleaching and Potential Transgenerational Effects

*Berghia stephanieae* is an aeolid nudibranch gastropod mollusc, that preys upon glass anemones that host photosynthetic endosymbionts and holds particular economic and scientific interest [35]. This nudibranch is a simultaneous hermaphrodite that is able to lay up to five spiral egg masses with hundreds of embryos per week. From 10 to 12 days after oviposition (depending on water temperature), either lecithotrophic larvae that undergo metamorphosis without feeding or imago of adults can hatch. Should the embryos hatch as larvae, metamorphosis can occur as soon as only one day post hatching. Exogenous feeding begins three to four days post-metamorphosis and sexual maturity can be reached as early as 47 days since oviposition [36,37]. More recently, studies have shown that, even though these nudibranchs are simultaneous hermaphrodites, they develop functional male gonads earlier than they do female ones, having the ability to store functional sperm for several months [38].

The rearing of this nudibranch in the laboratory has been well documented since the 1990’s [36]. The cultivation protocol is straightforward, although there are several adaptations/variations of the approach here being described. Briefly, artificial sea water, previously “seasoned” in an already running aquarium, filtered through a 0.45 µm filter should be used for culture. Adult broodstock can be maintained in small 600 mL finger bowls, with daily feeding of glass anemones, along with bowl and water changes. As egg masses are produced by adults, they should be collected and placed in 500 mL beakers with 350 mL of the filtered “seasoned” sea water and aeration. Beaker and water should be changed daily. After hatching, the water may be sifted through a Nitex filter to concentrate the larvae that should be subsequently placed in crystallizing dishes with filtered “seasoned” sea water and some very small glass anemones. After five to 10 days, daily water changes should begin, with change of bowl as needed. Approximately two to three weeks after metamorphosis, juveniles should be large enough to be transferred to finger bowls, where they can be reared as the adult broodstock [36].

*Berghia stephanieae* is not only a highly priced ornamental species for reef aquariums [39], it has also been advocated as a suitable model for the understanding of the molecular evolution of photosymbiosis in animals [16,40], as well as nematocyst sequestration [41] and neurodevelopment [42]. As a matter of fact, its ease of maintenance, regular oviposition, short life cycle and feeding upon a prey that can be readily cultured in the laboratory, are some of the key features that make *B. stephanieae* a remarkable model for laboratory research [36]. The scientific interest in this sea slug prompt researchers to determine its complete mitochondrial genome [43] and develop a rapid non-invasive technique using chlorophyll *a* fluorescence to monitor its association to the photosynthetic dinoflagellate endosymbionts it acquires from its prey [17].

The highly specialized diet of this nudibranch is most likely the trait that makes it more vulnerable to climate change, as *B. stephanieae* can most likely suffer the harshest consequences of bleaching [13,14,17]. To determine whether stenophagous animals are either “winners or losers” in the landscape of climate change, we need to better understand the consequences of bleaching on both predator (*B. stephanieae*) and prey (*E. diaphana*).

*Exaiptasia diaphana* is a widely used model species, with a reference transcriptome of adult aposymbiotic specimens being available since 2012 [44] and its genome having been fully sequenced in 2015 [45]. More recently, the microbiota of this anemone has also been characterized [46]. Even though this organism is already praised as, perhaps, one of the best models to study cnidarian-dinoflagellate symbiosis, there is still room for improvement, such as developing axenic (germ free) and gnotobiotic (with a known microbial community) culture protocols for *E. diaphana*. At present, there are at least three different strains of *E. diaphana* mainly used as model species for coral research: strain H2, from Coconut Island, Hawaii; strain CC7, from the South Atlantic Ocean, North Carolina; and strain RS from the Red Sea, Al Lith, Saudi Arabia [18].

With both *B. stephanieae* and *E. diaphana* being considered as suitable model species for multiple research fields, we advocate the use of this model pair to study the trophic impacts of bleaching induced by climate change in stenophagous organisms preying upon species hosting photosynthetic endosymbionts. However, caution should be employed as, even though the full mitochondrial genome of the sea slug has already been sequenced [43], there is still no full genome or even transcriptome available for *B. stephanieae*. Nevertheless, this opens up the opportunity for future efforts to sequence the genome of this popular nudibranch, as well as to determine its transcriptome, which would help to better understand its biology and physiology. The nutritional impact of bleaching in this sea slug, mainly on its fatty acid profile, has already been documented [13], and the ability to easily produce monoclonal populations of its prey (glass anemones) to better control laboratory experiments has also been described [47]. Hence, this model predator–prey pair certainly holds great potential for research on the trophic impacts of bleaching.

The advantageous traits displayed by *B. stephanieae* (i.e., short-life cycle and high egg production rate), coupled to well-established and simple laboratory rearing protocols [36], present an opportunity to develop new research avenues within global change biology and trophic interactions. In this way, we advocate *B. stephanieae* as a suitable model to assess the potential existence of transgenerational effects promoted by a suboptimal nutritional regime caused by bleaching of its prey. Indeed, the possibility to have both adults, egg masses, juveniles, and the prey organism in the same facilities (Figure 1), has already been used to study these carry-over effects, from parental *B. stephanieae* organisms to their egg masses [13]. Transgenerational plasticity, which occurs when parental environment influences the phenotype of their offspring (i.e., plasticity that occurs across generations) [48], has been the focal point of many studies testing responses to different physical and chemical changes in the environment [49]. However, even though both marine invertebrates and fish have been used on several of such studies, they were mainly focused on environmental changes and not trophic interactions [50,51].

## 4. Present and Future Uses for the Model Pair *Berghia stephanieae* and *Exaiptasia diaphana*

The bleaching of cnidarians hosting photosynthetic endosymbionts, namely corals, has been thoroughly investigated in the last 25 years [20,23,52,53,54]. The model-pair *B. stephanieae/E. diaphana* may help to advance the state of the art by making possible to study the effect of bleached and healthy symbiotic prey on a highly specialized predator, as well as on their offspring.

By articulating the study of this model-pair with prediction models for future climate scenarios, one may start to better understand the trophic effects promoted by bleaching on highly specialized predators, as well as the cascading effects of a parental suboptimal diet resulting from feeding on bleached prey on subsequent generations. Additionally, the multivariate dimension of global changes (not only temperature rises but also ocean acidification and hypoxia, which may synergistically contribute to bleaching) can be explored using this model predator–prey pair. Such studies will provide new insights on the true dimension that bleaching events may have on tropical coral reefs (Figure 2).

While *B. stephanieae* do not establish a true mutualistic relationship with ingested Symbiodiniaceae, being able to complete their life cycle by preying upon bleached glass anemones, new evidence suggests that feeding on bleached prey has negative impacts on reproductive fitness and juvenile growth [13], personal observation. Therefore, the trophic consequences of cnidarian bleaching on organisms that prey upon them (but do not establish short- or long-term associations with photosynthetic endosymbionts) should not be assumed to be negligible. On the contrary, such trophic consequences are more substantial than previously thought. We thus conclude that the model predator–prey pair here advocated will pave the way to study the plasticity and evolution of trophic interactions under global change scenarios, shedding light on how global change induced bleaching may shape the fate of highly specialized stenophagous animals from molecules to ecosystems, within and across generations.

## Figures and Tables

**Figure 1 animals-13-00291-f001:**
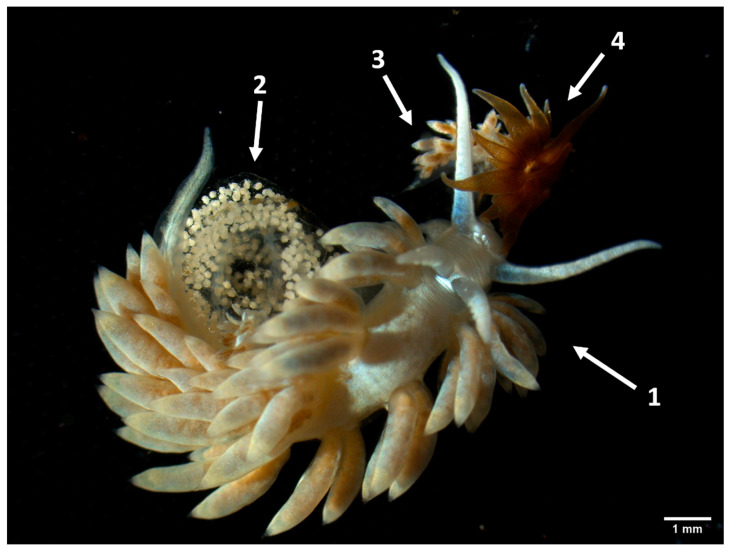
*Berghia stephanieae* adult (1), after laying eggs (2), and juveniles (3) feeding upon *Exaiptasia diaphana* (4).

**Figure 2 animals-13-00291-f002:**
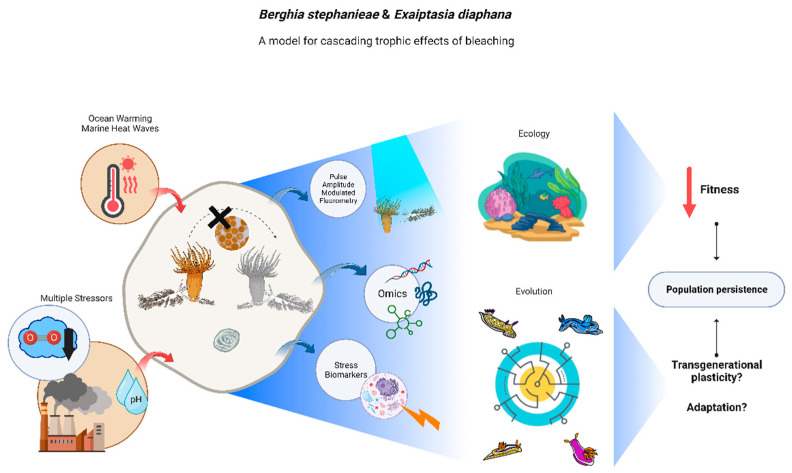
Graphical representation of the possibilities opened-up by the model-pair *Berghia stephanieae/Exaiptasia diaphana* for research on cascading trophic effects of bleaching, from molecular to ecosystem level. Blue arrows represent available methodologies to advance the state of the art and red arrows indicate environmental stressors to be addressed. Central part of the figure displays the different conditions on which one can study this model-pair: sea slug adults and juveniles feeding on symbiotic anemones and sea slug adults and juveniles feeding upon bleached anemones, and their egg masses. The right part of the image displays expected outcomes from this subpar feeding regime.

## Data Availability

Not applicable.

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
