# Peer review of "Assessing the Trophic Impact of Bleaching: The Model Pair Berghia stephanieae/Exaiptasia diaphana"

_animals, 2023, doi:10.3390/ani13020291_

Round 1

Reviewer 1 Report

Assessing the trophic impact of bleaching: The model pair Berghia stephanieae/Exaiptasia diaphana by Silva et. al., presents an interesting idea to study the trophic impact of bleaching by studying prey-predator interaction using Berghia stephanieae - Exaiptasia diaphana as a model pair. It has become very critical to understand the functional losses incurred in trophic interactions that are impacted by changing climatic parameters.

However, some studies have reported that Aeolid sea slug Berghia stephanieae do not establish a mutualistic relationship with the ingested Symbiodiniaceae and its growth did not depend upon the presence or absence of symbionts in the prey (Berghia stephanieae).

I suggest the authors to make a compelling case to use Berghia stephanieae here as a model predator instead of other aoelid sea slugs which comparatively have much higher retention time of symbionts like Phyllodesmium colemani or species which are known to establish mutualism with Symbiodiniaceae like Pteraeolidia ianthina. The schematics in figure 2 can be improved for more clarity, as to what is expected in terms of functional losses or deficient trophic interactions and larger ecological consequences.

There are few grammatical errors or missing references on line 62, 85 and 98. Furthermore if it is a review, probably, the article should have more comprehension. A revision should highlight the choice of model organism and will strengthen the case authors want to put forth. Since it is a very short text (with a suggested idea) and less literature available, it can be better suited as a Commentary or an Opinion article. I would suggest a major revision so as to consider for publication.

Reviewer 2 Report

The submitted review focused on the effects of climate change and the beaching on trophic interactions between the stenophagous nudibranch species B. stephanieae and the anemone E. diaphana hosting photosynthetic endosymbionts and the interesting proposal of these two marine invertebrates as a novel model predator – prey pair.

Moreover, this model pair could be useful allowing to gain in the future new insights on potential transgenerational effects promoted by the trophic impacts of bleaching on both predator and prey.

The review is well written and organized and liked it a lot.

I think that this review opens interesting new insights on a poorly investigated field by proposing a new model of study which involves however two well know and studied organisms showing ecological and biological characteristics that make them very useful for in situ and in laboratory studies.

For this reason, I think that this review should be considered for publication after minor revision.

In fact, after a deep revision I found no significant problems and my minor corrections and/or suggestions are directly reported here.

Line 84

Please add authors and year the first time you cite a scientific name.

Line 85

Please the scientific name should be in italic and with authors and year when you cite it for the first time.

Line 87

Maybe it is more correct ‘produces’ than ‘produce’, but I am not an English mother tongue.

Lines 95-98

A reference should be given.

Line 127

After ‘Coker and colleagues’ you should add the reference number.

Line 132

Is there a reference supporting this sentence? If yes it should be added here.

Line 134

Please add author and year to both the cited species.

Line 137

‘the anemone it lives in’ maybe better ‘the anemone it lives on’

Line 143-144

Please add author and year to both the cited species.

Lines 147 and 149

Please add author and year to both the cited species.

Line 151

Please add author and year to the cited species.

Lines 154

Please add author and year to all the cited genera.

Line 162

For the sake of clarity it should be added the class and the phylum to which nudibranchs belong.

Line 172

Please change ‘that is to able lay’ in ‘that is able to lay’

Lines 178-180

Sentence not clear, please correct or rephrase it. However, if I understood well, this species is not a simultaneous hermaphrodite as stated in line 172…

Line 200

Please correct ‘remarkabable’ with ‘remarkable’

Figure 1

Numbers 2 and 3 in Figure 1 are inverted.

Figure 2

My compliments for this beautiful figure!

Lines 275-276

Just a small suggestion…maybe it is better ‘feeding’ than ‘fed’?
